# Laboratory Calibration and Performance Evaluation of Low-Cost Capacitive and Very Low-Cost Resistive Soil Moisture Sensors

**DOI:** 10.3390/s20020363

**Published:** 2020-01-08

**Authors:** Soham Adla, Neeraj Kumar Rai, Sri Harsha Karumanchi, Shivam Tripathi, Markus Disse, Saket Pande

**Affiliations:** 1Chair of Hydrology and River Basin Management, Technical University of Munich, 80333 Munich, Germany; markus.disse@tum.de; 2Kritsnam Technologies Private Limited, Kanpur 208016, India; neeraj.rai@kritsnam.in (N.K.R.); harsha@kritsnam.in (S.H.K.); 3Department of Civil Engineering, Indian Institute of Technology Kanpur, Kanpur 208016, India; shiva@iitk.ac.in; 4Department of Water Management, Delft University of Technology, 2628 CN Delft, The Netherlands; s.pande@tudelft.nl

**Keywords:** volumetric water content, soil moisture, permittivity, capacitive sensor, SM100 sensor, SMEC300 sensor, resistive sensor, off-the-shelf sensor, calibration, temperature sensitivity, salinity dependence, low-cost sensor, irrigation management, precision agriculture

## Abstract

Soil volumetric water content (VWC) is a vital parameter to understand several ecohydrological and environmental processes. Its cost-effective measurement can potentially drive various technological tools to promote data-driven sustainable agriculture through supplemental irrigation solutions, the lack of which has contributed to severe agricultural distress, particularly for smallholder farmers. The cost of commercially available VWC sensors varies over four orders of magnitude. A laboratory study characterizing and testing sensors from this wide range of cost categories, which is a prerequisite to explore their applicability for irrigation management, has not been conducted. Within this context, two low-cost capacitive sensors—SMEC300 and SM100—manufactured by Spectrum Technologies Inc. (Aurora, IL, USA), and two very low-cost resistive sensors—the Soil Hygrometer Detection Module Soil Moisture Sensor (YL100) by Electronicfans and the Generic Soil Moisture Sensor Module (YL69) by KitsGuru—were tested for performance in laboratory conditions. Each sensor was calibrated in different repacked soils, and tested to evaluate accuracy, precision and sensitivity to variations in temperature and salinity. The capacitive sensors were additionally tested for their performance in liquids of known dielectric constants, and a comparative analysis of the calibration equations developed in-house and provided by the manufacturer was carried out. The value for money of the sensors is reflected in their precision performance, i.e., the precision performance largely follows sensor costs. The other aspects of sensor performance do not necessarily follow sensor costs. The low-cost capacitive sensors were more accurate than manufacturer specifications, and could match the performance of the secondary standard sensor, after soil specific calibration. SMEC300 is accurate (MAE, RMSE, and RAE of 2.12%, 2.88% and 0.28 respectively), precise, and performed well considering its price as well as multi-purpose sensing capabilities. The less-expensive SM100 sensor had a better accuracy (MAE, RMSE, and RAE of 1.67%, 2.36% and 0.21 respectively) but poorer precision than the SMEC300. However, it was established as a robust, field ready, low-cost sensor due to its more consistent performance in soils (particularly the field soil) and superior performance in fluids. Both the capacitive sensors responded reasonably to variations in temperature and salinity conditions. Though the resistive sensors were less accurate and precise compared to the capacitive sensors, they performed well considering their cost category. The YL100 was more accurate (MAE, RMSE, and RAE of 3.51%, 5.21% and 0.37 respectively) than YL69 (MAE, RMSE, and RAE of 4.13%, 5.54%, and 0.41, respectively). However, YL69 outperformed YL100 in terms of precision, and response to temperature and salinity variations, to emerge as a more robust resistive sensor. These very low-cost sensors may be used in combination with more accurate sensors to better characterize the spatiotemporal variability of field scale soil moisture. The laboratory characterization conducted in this study is a prerequisite to estimate the effect of low- and very low-cost sensor measurements on the efficiency of soil moisture based irrigation scheduling systems.

## 1. Introduction

The gravimetric method [1], which is the most accurate method of VWC measurement, is destructive, laborious, and does not provide results in real-time [2]. This has led to the development of nondestructive, indirect methods for the measurement of VWC [3,4,5,6,7,8]. Examples include neutron thermalization [9], time domain reflectometry (TDR) [10,11], time domain transmission (TDT) (e.g., in [12]), electrical capacitance [13,14,15] and impedance sensors (e.g., in [15,16]).

The electromagnetic (EM) sensors (TDR, TDT, and capacitance sensors) work on the principle that the EM wave propagation in bulk soil is primarily governed by liquid water that has a substantially larger dielectric permittivity ($\epsilon_{r}$) than the other soil components (gaseous air and solid soil minerals) [3]. TDR and TDT sensors operate at higher frequencies (of the order of GHz [12]) at which VWC measurements are less sensitive to soil electrical conductivity and imaginary dielectric permittivity [12]. Though the TDR method is regarded as the most accurate EM based VWC measurement technique [11,17], it is limited by its high cost and complex waveform analysis [2]. Capacitance and frequency sensors, developed as alternatives to the TDR technique [18], operate between 50 and 150 MHz [3]. They are similar with respect to repeatability, applicability to a wide range of soil types, and continuous monitoring ability [19], but are further advantageous due to significantly lower costs.

The combination of new technologies, stakeholders-cooperation and effective pro-poor institutions, within a larger and enabling policy framework, is considered to be the ‘best chance for lasting and sustainable impact on poverty’ [20]. These factors can result in an improvement of the livelihoods of the the poorest smallholder farmers through a transition towards sustainable agriculture [20,21]. The lack of supplemental irrigation facilities has been identified as a major exacerbating factor for smallholder farmers facing severe agricultural distress [22,23]. Therefore, there is a need for sensor based systems for VWC monitoring for applications such as irrigation management (for instance, through wireless sensor networks (WSNs)) [24]. The utilization of such technologies is challenged by low awareness and reluctance towards adoption in farmers, and a lack of interest in investment due to the economic pressures of fast returns on investments, which could be countered by developing low-cost and user-friendly systems [21].

Soil moisture sensors operating within WSNs can serve various purposes extending beyond irrigation management, such as validating remotely sensed soil moisture products [25], observing ecohydrological processes [26,27], or characterizing spatial soil properties [28,29]. However, WSNs are claimed to be expensive and to require further development [30]. To maximize the number of sensor nodes and due to the substantial quantity of VWC measurements from such networks, it is essential to use low-cost sensors with signals which can be interpreted in a straightforward and clear manner [3]. This has triggered an increase in the number of low-cost VWC sensors operating within WSNs spread over larger areas [3]. The cost of commercially available VWC sensors varies over four orders of magnitude. Capacitance sensors, being comparatively inexpensive and easy-to-use, show promise in measuring VWC within WSNs [31,32,33]. However, low-cost sensors may exhibit sensor-to-sensor variability [34], which, if not addressed, affects measurement accuracy [3]. In this study, two cost categories are defined: ‘low-cost’ and ‘very low-cost’. These categories are, respectively, approximately one order and three orders of magnitude lower than an expensive, TDR-based sensor (without considering data logger or reader costs).

Table A1 and Table A2 list some representative studies from a large body of work during the past few decades that have focused on the calibration and testing of VWC sensors. A list of publications, which have calibrated different capacitance, Frequency Domain Reflectometry (FDR), or impedance soil moisture sensors on various fluids or porous media, using different curve-fitting methods, is given in Table A1. The determination of sensor accuracy, precision, sensor-to-sensor variability, volume of influence, and temperature and salinity effects is also vital to understand sensor performance under different conditions encountered in practice (some publications listed in Table A2). The novelty of this study lies in characterizing and testing non-research grade soil moisture sensors (in particular the low-cost capacitive and very-low cost resistive sensors), which is a prerequisite to assess their irrigation management capabilities. Also, a new approach for holistic visualization of sensor accuracy and precision for multiple sensors and soils is presented.

Partly based on the literature [35], and partly motivated by the impact that low-cost soil moisture sensors could have on ecological research and supplemental irrigation, the following questions were used to design the experiments conducted in the study.

What is the ability of the capacitive sensors to estimate the refractive index ($\epsilon_{r}$) of various fluids of known $\epsilon_{r}$ values?What empirical equation(s) can best explain the relationships between the output of the low- and very low-cost soil moisture sensor instruments tested in the study, and the actual VWC, across a variety of soils?What is the difference between the respective accuracies of the soil-specific calibration equations developed in-house and the general manufacturer-provided calibration equations?What is the accuracy and precision performance of different low- and very low-cost soil moisture sensor instruments tested?How is the accuracy and precision of the developed calibration curves affected by variations in (i) temperature and (ii) electrical conductivity, within ranges that are commonly encountered in field conditions?

The results pertaining to the above questions are addressed in Section 3.1.1, Section 3.1.2 and Section 3.1.3, Section 3.2 and Section 3.3, respectively.

## 2. Materials and Methods

### 2.1. Soil Moisture Sensors

The sensors tested in the study are described in the subsections below. A comparison of the salient features (including prices from quotations) and the corresponding cost-based nomenclature used in the study are presented in Table 1. Sensor photographs are given in Figure 1.

#### 2.1.1. Capacitance Based Low-Cost Sensors: Spectrum SM100 and SMEC300

The WaterScout SM100 Soil Moisture Sensor (manufactured by Spectrum Technologies, Inc., Plainfield, IL, USA) is a capacitance-based low-cost soil moisture sensor [36]. The sensor has a pair of electrodes that operates as a capacitor and the surrounding soil functions as the charge storing dielectric medium [36]. The WaterScout SMEC 300 Soil Moisture Sensor is also a capacitance-based low-cost soil moisture sensor with the additional capability of measuring EC and soil temperature [37]. In both cases, an oscillator operating at 80 MHz drives the capacitor and the generated output (voltage ratio) is proportional to the soil’s dielectric constant ($\epsilon_{r}$) [36]. However, the estimated $\epsilon_{r}$ is not available to the user via data loggers or readers as both sensors are calibrated by developing relationships between voltage ratios/raw A/D (analog to digital) values and actual VWC (θ) of a continuously drying soil column (D. Kieffer, personal communication, 5 September 2018). Additionally, the SMEC300 sensor measures EC with a pair of carbon ink electrodes, and temperature using a thermistor potted in the sensor molding [37]. The SM100 has a reported accuracy of 3% VWC at an EC <800 mS·m^−1^, and an operating range of 0.5 °C to 80 °C [36]. The SMEC300 has reported accuracies of 3% for VWC, ±1 mS·m^−1^ for EC and 0.6 °C (0.8 °C) for temperatures greater than −30 °C (lesser than −30 °C), and has ranges of operations of 0–1000 mS·m^−1^ for EC and −50 °C to 85 °C for temperature [37].

#### 2.1.2. Generic Resistance Based Very Low-Cost Sensors: YL100 and YL69

The Soil Hygrometer Detection Module Soil Moisture Sensor provided by Electronicfans (herein referred to as YL100) and the Generic Soil Moisture Sensor Module by KitsGuru (herein referred to as YL69) are both resistive soil moisture sensors. Both the sensors have two pronged probes operating as variable resistances which are a function of the soil moisture. An increasing soil moisture increases the effective conductivity of soil [38,39]. This variation in resistance causes a variation in voltage drop, which is then measured by the electronic module and subsequently returned as an output. However, the measured soil resistivity is also influenced by ion concentration [40] and hence careful calibration along with frequent recalibrations (due to variable organic and salt concentrations) are recommended for effective application [41]. Previous studies have developed calibration curves for estimating continuous soil moisture [38] or soil moisture categories (dry, medium, high, etc.) [39] as a compromise between sensor accuracy and cost. No records of sensor specifications (including accuracy and operating conditions) could be found in the literature for either of the sensors.

#### 2.1.3. Impedance-Based Sensor: Delta-T ThetaProbe ML3

The Delta-T ThetaProbe ML3 (henceforth referred to as the ThetaProbe) measures the soil VWC by responding to the changes in its apparent dielectric constant [42]. A 100 MHz sinusoidal signal is applied to an internal transmission line extending into the soil by means of a sensing head [42]. This comprises of an array of four rods: three of them (connected to the instrument ground) behaving as an electrical shield around the central, signal rod. The sensing head operates as an additional section of transmission line and has an impedance which depends on the dielectric constant of the soil [43]. The impedance of the rod array subsequently impacts the reflection of the 100 MHz signal at the junction between the internal transmission line and the sensing head [42] and the interference of the reflected component with the incident signal causes a standing wave to form on the transmission line [43]. The output is an analog voltage proportional to the difference in amplitude of this standing wave at two points—the junction and the starting point of the transmission line [42]. This amplitude is related to the relative impedance of the probe, and thus the dielectric constant and VWC [43]. The VWC sensor has a two-step calibration process; a soil specific linear calibration equation between the actual VWC (θ) and the refractive index ($\sqrt[{}]{\epsilon_{r}}$) of the dielectric medium, and a sensor specific 6th degree polynomial calibration equation between the output voltage and the refractive index ($\sqrt[{}]{\epsilon_{r}}$), together resulting in a 6th degree polynomial calibration equation between the output voltage and the actual VWC (θ) [44]. The ThetaProbe has a reported soil moisture accuracy of ±1%, salinity error of ≤3.5% VWC over 50–500 mS·m^−1^ and 0–50% VWC, and soil temperature accuracy of ±0.5 °C over 0 °C to 40 °C [44]. It is considered to provide a sensitive and precise measurement of VWC and soil temperature [44], and is accepted for surface soil water content measurements [24]. Therefore, it could be justified to be used as a secondary standard [45] for the different experiments conducted in this study.

### 2.2. Description of the Soils Used

The four different soils used in the study are shown in Figure 2, and a description of their physical characteristics is tabulated in Table 2. These consisted of two Indian Standard sands from IS 650:1991 [46] and two silty-loam soils representative of agricultural landuse in the Ganga floodplains. Among the silty-loams, Soil 3 was sampled from a local agricultural field and included without any grading (to purposefully represent local field conditions), as opposed to Soil 4, which was graded with a 2 mm sieve.

### 2.3. Sensor Calibration

#### 2.3.1. Calibration of Capacitive Sensors with Fluids

Following the literature [2,3,48,49], fluids of known dielectric properties were used to evaluate (i) sensor accuracy, (ii) sensor precision, and (iii) the comparative performance of the tested capacitive sensors. The motivation behind using fluids was to minimize the variability in measurements arising due to nonuniform contact between the sensor surface and the porous media [2]. The fluids were chosen because their respective $\epsilon_{r}$ values were known and those values fall in the range generally encountered in soils of varying VWCs. The fluids selected for the study are shown in Table 3. The deionized water is henceforth referred to as “water”.

Note that as the ThetaProbe was taken to be a secondary standard, it was included as a standard against which the investigated sensors were tested, rather than being calibrated or tested for performance.

#### 2.3.2. Calibration of Sensors with Repacked Soils

In addition to calibrating soil moisture sensors with fluids of different $\epsilon_{r}$ values (which was relevant only for the capacitive sensors), it is essential to calibrate sensors in porous media (such as repacked or natural soils) before effective field application. Although repacking alters the natural soil structure [35], using repacked soils for calibration is recommended to achieve better precision [50]. The calibration methodology adopted was based on predetermined uniform soil water content regimes for repacked soils, similar to recent studies [24,49]. Known quantities of water were added to containers with oven dried soils. The actual VWC (θ) was determined using a weighing machine, and multiple VWC measurements ($\hat{\theta_{i}}$) were taken with multiple specimens of each of the four sensors tested in the study. This process was repeated for each of the four soils (Table 2). The calibration methodology used for the repacked soils is described in detail on an online database, https://www.protocols.io/ [47], to encourage methodological reproducibility and refinement.

### 2.4. Performance Measures for the Sensors

#### 2.4.1. Sensor Accuracy

Accuracy is a measure of how close the measured output is to the true value [51]. Accuracy may also be defined as the maximum difference that exists between a measured value and the true value determined by a standard reference procedure [52]. In this study, the true value was determined through two approaches: a primary calibration standard (the gravimetric weight) as well as a secondary calibration standard (the impedance-based ThetaProbe soil moisture sensor). Three measures were used to quantify accuracy: mean Absolute Error (MAE), Root Mean Squared Error (RMSE), and Relative Absolute Error (RAE, σ). They are described, along with other performance measures, in Section 3.

#### 2.4.2. Sensor Precision

Precision describes a measurement’s repeatability, which indicates the extent to which consecutive measurements of the same input produce the same output [53]. In this study, precision was defined using the Pooled relative standard deviation ($s_{r,p}$), which provides an overall estimate of imprecision by combining the standard deviations around the respective means across a series of measurements [54]. The multiple series of measurements corresponded to different mean values of measured VWC (for instance, $\bar{\hat{\theta_{m}}}$ for the *m*th series), and the $s_{r,p}$ is defined in Section 3.

### 2.5. Sensor Sensitivity

#### 2.5.1. Temperature Sensitivity

Multiple studies have investigated the effect of ambient and soil temperature on VWC measurements [2,35,55,56,57]. Further methods have been proposed for correcting errors in VWC measurements arising due to diurnal variations in temperature [58]. For this study, capacitive and resistive soil moisture sensors were tested in a silty-loam soil (Soil-4, described in Table 2) with two different values of actual VWC and ambient temperatures ranging from 10 °C to 40 °C (with an error of ±1 °C), inside a temperature incubator. The soil surfaces were covered with polythene sheets to prevent evaporation. To ensure that the electronic components of the sensors (excluding the sensing element which is inserted in the soil) were not affected by the temperature variations, they were placed outside the incubator.

#### 2.5.2. Salinity Sensitivity

The dependence of the measured VWC on salinity was determined following the method suggested in the literature [2,49]. Varying amounts of water (to cover a range from dry to saturation) with known KCl concentrations were added to Grade III sand (Soil-2, Indian standard [46] described in Table 2) and VWC measurements were made using the different sensors. A total of 12 samples, as described in Table 4, were studied.

## 3. Results and Discussion

The performance measures used in developing the results in the study are listed in Table 5 along with their respective sources from the literature.

### 3.1. Sensor Calibration

#### 3.1.1. Performance of Capacitive Sensors with Fluids

The capacitive sensors SM100 and SMEC300 were first tested with fluids of known $\epsilon_{r}$ (Table 3). The secondary standard, ThetaProbe, provides an estimate of the Refractive Index ($\sqrt[{}]{\epsilon_{r}}$), whereas the capacitive sensors do not directly measure $\epsilon_{r}$ but relate raw sensor output to VWC. Therefore, first, the VWC values of the capacitive sensors were converted to $\sqrt[{}]{\epsilon_{r}}$ values using the expression given by Topp et al. [10]:
$$\theta =-5.30\times 10^{-2}+2.92\times 10^{-2}\epsilon_{r}-5.50\times 10^{-4}\epsilon_{r}^{2}+4.30\times 10^{-6}\epsilon_{r}^{3}$$
where:

θ = VWC (%)

$\epsilon_{r}$ = Dielectric constant (-)

Further, they was compared with the respective actual values and $\sqrt[{}]{\epsilon_{r}}$ values measured by the ThetaProbe. The results of this analysis are provided in Figure 3 and Table 6. The estimated $\sqrt[{}]{\epsilon_{r}}$ values (along with their standard errors) for all the three sensors are depicted on the Y-axis, and the actual $\sqrt{\epsilon_{r}}$ (derived from known $\epsilon_{r}$ values at 25 °C) are plotted on the X-axis of Figure 3.

The ThetaProbe sensor has relatively good performance in measuring refractive indices of air, Butanol and Ethanol compared to ethylene glycol and water. The precision values (standard deviations) of the ThetaProbe in measuring $\sqrt[{}]{\epsilon_{r}}$ in air, butanol, ethanol, ethylene glycol, and water were SD = 0.003, 0.053, 0.156, 0.401, and 0.621, respectively. The overall precision of the ThetaProbe was $s_{r,p}$ = 0.0405.

Figure 3 indicates that the ThetaProbe sensor was more accurate than both the SMEC300 and SM100 sensors in all the fluids except Ethylene Glycol. Both SMEC300 and SM100 have equal precision values when considered till the fourth decimal place ($s_{r,p}$ = 0.0062), and are more precise when compared to ThetaProbe.

The comparison of capacitive sensors in fluids suggests that both the sensors were equally precise; however, the inexpensive SM100 sensor outperformed the SMEC300 sensor in terms of MAE and RMSE. Overall, these results were encouraging realizations of the sensing abilities of both the capacitive sensors in general, and the SM100 sensor in particular, under the conditions of uniform contact between the sensor and the dielectric medium.

#### 3.1.2. Calibration of All Sensors with Repacked Soils

##### Strength of Monotonic Relationship Between Measured ($\hat{\theta }$) and Actual (θ) VWC

The Spearman’s rank correlation coefficient, $r_{s}$ [61], was employed to assess the strength of the relationship between $\hat{\theta }$ and θ. The Spearman’s rank correlation was selected because it is a nonparametric statistic that measures the strength of a monotonic relationship between paired data without any assumptions made on the distribution of the data or the nature of the relationship existing between them [62,63,64]. The number of sensor units of the SMEC300, SM100, YL100 and YL69 sensors used for the experiment were 6, 5, 6, and 5, respectively. Table 7 illustrates the $r_{s}$ values for the four different sensors tested in the study. For all the sensors and soils, the Spearman’s rank correlation was positive and significant at the 5% level.

The capacitive and the resistive sensors had an average $r_{s,resistive}$ = 0.93 and 0.87, respectively, averaged across all soils. Among the capacitive sensors, the SM100 sensor ($r_{s}=0.94$) performed better than the SMEC300 sensor ($r_{s}=0.92$) on average. For each soil, both the sensors had roughly same $r_{s}$ (within 1 to 3% of each other) except for Soil 3, in which SM100 ($r_{s}=0.94$) substantially outperformed SMEC300 ($r_{s}=0.84$). This difference may be attributed to soil characteristics, which was an ungraded field soil purposefully included in the study to represent local field conditions. These results implied that SM100 could outperform the SMEC300 sensor as a robust, field ready capacitive sensor (on the basis of VWC measures in repacked soils). These results advanced the results obtained in Section 3.1.1, where the SM100 sensor outperformed the SMEC300 sensor in fluids. Among the resistive sensors, the YL100 sensor ($r_{s}=0.89$) performed marginally better on average, compared to the YL69 sensor ($r_{s}=0.86$). The YL69 sensor performed better for both the sandy soils Soil 1 (by 16.7%) and Soil 2 (by 5.62%), compared to YL100. However, for the silty-loam soils, YL100 performed better ($r_{s}=0.94$ in both cases) than YL69, which itself performed worse in Soil 3 ($r_{s}=0.73$) than in Soil 4 ($r_{s}=0.85$).

Overall, considering only rank correlations, the order of performance was SM100 > SMEC300 > YL100 > YL69. This result could be expected in terms of the capacitive sensors being more accurate compared to the resistive sensors. To further strengthen these inferences, calibration equations were developed for each sensor.

##### Calibration Equations Developed between Measured ($\hat{\theta }$) and Actual (θ) VWC

The soil-specific calibration equations were piecewise linear regression equations based on the least squares estimate, in which the objective function, Sum of Squared Residuals $SSR=\sum_{i=1}^{n}{\left(\theta_{i}-\hat{\theta_{i}}\right)}^{2}$, was minimized. The number of line segments were decided using visual inspection, while an open source Python library, pwlf [65], was used to develop the corresponding piecewise linear equations.

The subsequent sections on sensor testing have used these calibration equations developed for each sensor and soil. Each segment of the piecewise linear equations, outlined in Table A3 and illustrated in Figure 4, are of the following form,
(1)$$\theta =\beta_{0}+\left(\beta_{1}\times \hat{\theta }\right)$$
where,

$\hat{\theta }$ = Raw sensor value (-),

θ = Actual VWC (%), and

$\beta_{i}$ = Calibration coefficients

The performance of the calibration equations developed for the capacitive sensors was comparable to that of other low-cost capacitive sensors reported in the literature [66]. The SM100 sensor (average overall $R^{2}=0.94$) performed at par with other low-cost capacitive sensors for sandy soils (average $R^{2}=0.95$ compared to $R^{2}=0.97$ from the literature [66]), and surpassed previous work for silty-loams (average $R^{2}=0.93$ compared to $R^{2}=0.88$ from the literature [66]). The SMEC300 performed equally well as the SM100 in sands (average $R^{2}=0.95$), but not in silty-loam soils (average $R^{2}=0.82$); nevertheless, being comparable to previous literature [66]. Overall, the calibration results reinforce the inference that the SM100 sensor is more robust (due to its superior performance in a field soil) compared to the SMEC300 sensor.

Understandably, the resistive sensors did not perform as well as the capacitive sensors. However, considering the fact that they were very-low cost sensors, the average performances of both the YL100 (average $R^{2}=0.81$) and the YL69 sensors (average $R^{2}=0.76$) were notable. Though a literature-based comparison to previous calibrations of low-cost resistive sensors was not possible, it emerged that the YL69 sensor performed reasonably well for sands (average $R^{2}=0.89$) and the YL100 performed well for silty-loam soils (average $R^{2}=0.85$).

Considering the extent to which the calibration equations could explain the variation in the measured data (through the $R^{2}$), the order of performance was SM100 > SMEC300 > YL100 > YL69. This is identical to the order of performance based on the Spearman’s rank correlation ($r_{s}$). These results are encouraging as all the sensors perform well compared to the results reported in the literature, where applicable.

#### 3.1.3. Comparison of Manufacturer and In-House Calibration Equations: Capacitive Sensors

Figure 5 compares the performance of the calibration equations developed in-house during the study and provided by the manufacturer, for the capacitance sensors (calibration equation were not available for the very low-cost resistive sensors). Additionally, Table 8 compares accuracy measures MAE, RMSE and RAE for manufacturer’s and in-house calibration equations for the four soils. The manufacturer calibration equations for the capacitance sensors were made available from Spectrum Technologies, Inc. (D. Kieffer, personal communication, 5 September 2018). The number of SMEC300 and SM100 sensors used for this experiment are six and five, respectively.

From Figure 5, it was observed that for both the sensors, the manufacturer’s equations had a tendency to underpredict the actual VWC (θ), with the exception of Soil 3 (ungraded silty-loam soil) at higher VWC (θ) values. Sensor accuracy increased substantially, overall as well as in each soil, after soil specific calibration equations were developed. From Table 8, it could be inferred that the sensor accuracy without calibration, for both the capacitive sensors, was lower than the accuracy specified by the manufacturer, 3% (for both SMEC300 [37] and SM100 [36]). This observation was in line with the claim that it is “optimistic” to expect such levels of accuracy for many EM sensors [67]. After soil specific calibrations, there were substantial improvements in sensor accuracy. The MAE (and RMSE) of the calibrated SMEC300 and SM100 sensors were 2.12% and 1.67% (2.88% and 2.36%) respectively, which were better than the manufacturer reported accuracy values.

It was hence evident that more effective overall performance can be ensured with soil-specific calibration equation development and installation based on the soil texture established in the field, which supports the previous literature [2]. This performance enhancement would ideally compensate for the resources (financial, human) incurred in the exercise.

### 3.2. Performance Measures for the Sensors

Figure 6 illustrates the accuracy ($\sigma_{eff}$) and precision ($s_{r,p}$) of the tested sensors together in a bubble plot in a 2-D Euclidean space. Although the primary accuracy ($\sigma_{primary}$) evaluates sensor performance with actual VWC values (θ), secondary accuracy ($\sigma_{secondary}$) compares the sensor to the ThetaProbe, which was considered as the secondary standard due to its superior measurement technique [44]. The relevant performance indicators for effective accuracy ($\sigma_{eff}$), component accuracies ($\sigma_{primary}$ and $\sigma_{secondary}$) and precision ($s_{r,p}$) are described in Table 5.

Each accuracy component is represented by the distance from the origin to the cross-hair centers of the bubbles in the respective directions ($\sigma_{primary}$ along the X-axis and $\sigma_{secondary}$ along the Y-axis). Therefore, the effective accuracy ($\sigma_{eff}$) can be described as the Euclidean distance of the cross-hair centers of the bubbles (i.e. the closer the bubble center from the origin, the more accurate the sensor is). As the $\sigma_{secondary}=0$ for the ThetaProbe as it is computed with respect to itself, $\sigma_{eff}=\sigma_{primary}$ and its accuracy is defined only by the distance of the cross-hairs along the X-axis direction. The performance of all the sensors in each soil is represented in different quadrants (i.e. each soil has a corresponding quadrant, labeled in the figure). Precision is represented by the radius of the bubble graphs, which are proportional to $s_{r,p}$; $radius=100\times s_{r,p}$). Therefore, the bubbles with smaller radii are more precise in their VWC estimates ($\hat{\theta }$).

#### 3.2.1. Sensor Accuracy

An analysis of metrics related to accuracy revealed the order of performance as SM100>SMEC300>YL100>YL69, which was identical to the results in Section 3.1.2. The metrics included Mean Absolute Error (MAE), Root Mean Squared Error (RMSE), and the primary Relative Absolute Error ($\sigma_{primary}$ values, reported in Table 8). As shown in Figure 6, the effective accuracy ($\sigma_{eff}$) computed after including the secondary standard sensor measurements, gave rise to a slightly different order of performance, i.e., SM100>SMEC300>YL69≳YL100. The capacitive sensors outperformed the resistive sensors by a factor of 2 on average ($MAE_{capacitive}=1.90\%$, $MAE_{resistive}=3.82\%$; $RMSE_{capacitive}=2.62\%$, $RMSE_{resistive}=5.38\%$; $RAE_{capacitive}$ or $\sigma_{capacitive,primary}=0.25$, $RAE_{resistive}$ or $\sigma_{resistive,primary}=0.39$).

Both capacitive sensors had accuracy measures comparable to the previous literature pertaining to low-cost capacitive sensors, in terms of MAE and RMSE [33,66]. In terms of RAE, the SMEC300 sensor was accurate across all soils (average $\sigma_{eff}$ = 0.47, Standard Error $SE_{\sigma_{eff}}$ = 0.07) barring Soil 3, in which the $\sigma_{eff}$ reduced to 0.69. The SM100 was also accurate across all the soil types (average $\sigma_{eff}$ = 0.43 with a lower $SE_{\sigma_{eff}}$ of 0.03). Comparatively, the SM100 sensor outperformed the SMEC300 sensor in terms of overall effective accuracy ($\sigma_{eff}$), but largely due to the substantially better performance in Soil 3, supporting the results in Section 3.1.2. The SMEC300 was more accurate than the SM100 in both the sands and in Soil 4.

The MAE and RMSE values of the resistive sensors could not be compared with the existing literature on resistive sensors, but were poorer than the specified accuracy values (3%) of most EM sensors [67]. In terms of RAE, their accuracy values were enhanced by 56% (average $\sigma_{primary}$ = 0.39, varying between 0.32 and 0.48) when only primary accuracies were considered, which implied that they were able to capture variations in actual VWC better than the variations taking into account the secondary standard VWC. Comparatively, though the overall accuracies of both the resistive sensors across all the soils were similar, it could be remarked that each sensor complemented the other’s performance in a particular soil texture category. YL100 performed better in silty-loam soils while YL69 performed better in sandy soils.

Overall, it can be remarked that the low-cost sensors, when calibrated, could match (or exceed) the accuracy performance of the secondary standard sensor. YL100 was ~10% less accurate than the secondary standard ThetaProbe in terms of MAE, while its RMSE, and both MAE and RMSE values of YL69 were poorer when compared to the ThetaProbe.

#### 3.2.2. Sensor Precision

The precision performance of each sensor across all four tested soils, along with the performance of the secondary standard sensor, is given in Table 9. The order of overall precision (averaged across all soils) nearly followed the order of the cost of the sensors, i.e., SMEC300>SM100>YL69>YL100 with the ThetaProbe being the most precise (as well as expensive) sensor ($s_{r,p}$ = 0.31). Capacitive sensors were about twice as precise as the resistive sensors ($s_{r,p(resistive)}=0.79$, $s_{r,p(capacitive)}=0.37$). Additionally, the lowest precision achieved by the capacitive sensors was only 18% poorer than the precision of the ThetaProbe.

Both the capacitive sensors had a consistently high precision across all soils. The SMEC300 had a higher precision overall than the SM100 and also across all soils barring Soil 3. However, the SM100 was more consistent in terms of its precision performance across the different soils ($SD_{s_{r,p,SM100}}$ = 0.11) when compared to SMEC300 ($SD_{s_{r,p,SMEC300}}$ = 0.21). The SMEC300 sensor was reasonably precise compared to the ThetaProbe in Soils 1, 2, and 4, and even exceeded the ThetaProbe performance in Soil 2 (in which it had a very high precision of $s_{r,p}$ = 0.05). It was more precise in the soils with the finer grained soils within the two categories (Soil 2 and Soil 4 in the sandy and silty-loam soils, respectively), which was an indicator of the need for better packing around the sensing material during installation. The SM100 sensor also had comparable precision performance vis-à-vis the ThetaProbe (it performed within 31.2% of the ThetaProbe on average). Its performance increased with better packing in the sandy soils (Soil 2 is 20% more precise than Soil 1), but it performed well in both the silty-loams, with its performance being more precise in the ungraded silty-loam field Soil 3 than that in Soil 4 by 16.7%. The consistent precision performance of SM100 across soils in general and in the field Soil 3 in particular suggests that SM100 is a robust and field ready sensor.

Both the resistive sensors were reasonably imprecise compared to the ThetaProbe. The YL69 sensor was a more precise sensor compared to YL100, both overall and also in each soil. YL69 was almost as precise as the capacitive sensors in Soil 3. This was also an encouraging result for field application. Otherwise, the YL69 is within 85.8% and 58.9% of the performance of the capacitive sensors in sandy and silt-loam soils, respectively. YL100 also had better performance (by 32.1%) in silt-loam soils compared to the sandy soils, but was equally imprecise in both the sands and silty-loam soils. It was quite imprecise compared to the ThetaProbe (3 times as imprecise) as well as the capacitive sensors (2.54 times as imprecise).

### 3.3. Sensor Sensitivity

#### 3.3.1. Temperature Sensitivity

The subfigures of Figure 7 show the results of the temperature sensitivity experiments conducted for all sensors in Soil 4 for two different values of actual VWC (θ), depicted by the dashed and solid horizontal lines. The hollow circles and filled squares represent the average VWC estimated by each sensor (based on the corresponding calibration equations in Table A3) along with their respective standard errors. The temperature variation is plotted for each sensor on the X-axis. The SMEC300, SM100, YL69 and YL100 sensors had, on average, 6, 211, 276, and 276 data points, respectively, for each VWC-temperature combination. The data from SM100, YL100 and YL69 sensors were automatically read using open source Arduino (https://www.arduino.cc/) electronics while the hand-held FieldScout soil sensor reader was used for SMEC300 readings. The lower number of readings for the SMEC300 was a result of the effort to minimize the number of times the incubator door was opened, to consequently lower the variation in the incubator temperature.

The capacitive sensors showed a positive temperature effect (larger temperature leading to a higher estimated VWC), which is characteristic of capacitance sensors in soils with fine textures due to the release of bound water from clay minerals at higher temperatures [2]. The SMEC300 sensor followed the positive temperature effect at the lower actual VWC of 9.36%, with an overall average estimated VWC change of 9.34% responding to a 36.8 °C temperature increase. However, though the temperature effect was not visible at the higher actual VWC of 21.39%. This is justified through the corresponding fitted calibration curve (Soil 4; indicated in dark brown in Figure 4a and Table A3) which flattens out at higher raw values (> 1525), and subsequently limits the increase in estimated VWC as a consequence of increasing raw values. Therefore, the positive temperature effect not being visible at the higher actual VWC value is due to the calibration curve and not the physical changes in the measurement. Similarly, the SM100 sensor followed an expected positive temperature effect, for both the levels of actual VWC. At the actual VWC values of 7.63% and 18.38%, respectively, an increase of 30 °C resulted in an increase of estimated VWC by 7% and 2.99%, respectively. The results suggest that the temperature response of the capacitive sensors SMEC300 and SM100 could be predicted and consequently corrected.

The resistive sensors responded in dissimilar manners to the change in temperature at different actual VWC conditions. The YL100 sensor showed a temperature effect which was seen across larger temperature ranges, but there was a relatively large amount of variability which was seen at smaller temperature differences. Since this variability in measurements dominated the temperature sensitivity, characterizing the temperature sensitivity was difficult in the case of YL100. The YL69 sensor responded with a positive temperature effect, which was less pronounced in the lower actual VWC compared to the higher actual VWC. The YL69 under-estimated the lower value of the actual VWC (7.32%) for all the temperatures. Though this was due to its low accuracy compared to the capacitive sensors and the resultant calibration equation, which underpredicted the actual lower VWC, there was actually a small increase of the estimated VWC by 0.36% over 30 °C. The sensor response to the higher actual VWC (of 17.89%) was substantially closer to what was expected, with a positive temperature effect translating to a rise in 9.33% over the same 30 °C rise in temperature. If the calibration equation was not used, the positive temperature effects were more clearly seen, with increases of 11.03 and 284.98 in the raw values corresponding to actual VWC values of 7.32% and 17.89%, over the same rise in temperatures. The overall behavior implied that notwithstanding the lower accuracy performance, the temperature response of the YL69 sensor was reasonable and would hence be possible to correct.

#### 3.3.2. Salinity Sensitivity

Figure 8 plots the calibrated VWC (based on the equations developed in Section 3.1.2, listed in Table A3) and the actual VWC with changing salinity in water. The corresponding EC values were 1.70, 3.02, 6.32, and 9.69 mS/cm, represented by circular, triangular, square, and pentagonal shapes, respectively.

Based on the coefficients of determination ($R^{2}$ values), the order of performance was SM100>SMEC300>YL69>YL100, i.e., SM100 results were least sensitive to changes in salinity. The corresponding $R^{2}$ values were 0.85, 0.79, 0.63 and 0.13, respectively. These results are encouraging as the SM100 sensor, despite being the relatively less-expensive capacitive sensor, outperformed the more expensive SMEC300 by 7.6%, and the very low-cost resistive YL69 was only 20.5% less effective compared to the performance of the SMEC300 in response to salinity variations. However, YL100 was almost completely unable to capture any variation, and estimates more or less the same value (SD = 0.57%) irrespective of the actual VWC or the EC of the added water. Additionally, the bulk soil EC values measured by the SMEC300 sensors (median values based on the manufacturer calibration) are plotted in Figure 8e. As expected, an increase in the EC of the added water led to an increase in the bulk soil EC measured by SMEC300. The best-fit which minimized SSR was significantly linear (at α=5%) and had an $R^{2}$ = 0.92.

However, these results were inferred from an experiment with the sandy Soil 2, and further testing would be necessary to extend the results for each of these sensors to more generalized applications.

### 3.4. Further Discussion

The drawbacks of the experiment included and were not limited to experimental and human errors, as well as the choice of piecewise linear calibration functions. Further, the possibilities to represent more natural variability (by incorporating more soils), and introducing the packing density as an experimental variable (which has shown to have an impact in similar experiments [24]) were not integrated due to resource constraints. Moreover, it was assumed that a laboratory characterization is an essential precursor to field trials and experiments, especially since such a laboratory study involving these particular sensors had not been conducted earlier. Another aspect which was not tested, either in the laboratory or field conditions, was the durability of the sensors.

Having quantified the operations of these sensors, a case can be developed, within the larger framework of low-cost technological tools in agricultural water management, to propose the use of such sensors based on an understanding of the required precision of the problem as well as harness the complementary strengths of the sensors in different aspects of performance. For example, within the resistive sensors, there was a differentiation among the sensors in accuracy estimates across the grain size of the soils (YL69 and YL100 respectively outperformed the other resistive sensor in coarser and finer grained soils respectively). Such a characterization is valuable in choosing the correct sensor for a particular application case.

The efficiency of soil moisture based irrigation scheduling systems is dependent strongly on the sensor accuracy, with 3% errors in soil moisture sensors possibly leading to ‘critical’ effects on irrigation efficiency [67]. Therefore, although the capacitive sensors tested in this study had accuracy levels (<2% VWC on average) possibly leading to ‘limited’ effects on irrigation efficiency, using resistive sensors independently (with an accuracy of <4% VWC on average) could have potentially critical effects. The actual effect of these sensors on irrigation water use efficiency can be determined with comprehensive field experiments.

Field scale soil moisture distribution exhibits high spatial and temporal variability [68]. Instead of a sparse network of capacitive sensors, a dense network combining capacitive and resistive sensors could help better characterize the spatiotemporal variability of soil moisture, which may potentially improve irrigation management. Such a characterization using a combination of low- and very-low cost soil moisture sensors has not been attempted, as per the knowledge of the authors.

## 4. Summary and Conclusions

Four soil moisture sensors, i.e., two low-cost capacitive sensors (SMEC300 and SM100) and two very low-cost resistive sensors (YL100 and YL69), were tested in laboratory conditions to characterize their performance for application in low-cost irrigation management. Based on the literature, five research questions were developed and addressed with specific laboratory experiments. Piecewise linear calibration equations were developed for each sensor in four different repacked soils to explain the relationship between sensor measurements and actual VWC values. In the case of the capacitive sensors, the manufacturer-provided calibration equations were compared with the calibration equations developed in-house in terms of accuracy measures. Such a comparative analysis could not be performed for the resistive sensors due to the unavailability of manufacturer calibration equations. An evaluation of sensor accuracy and precision was conducted for all the studied sensors in all the tested soils and a novel approach to visually represent the combined performance characteristics is proposed. The sensitivities of the sensors were evaluated for temperature and salinity ranges commonly encountered in field conditions. Additionally, only for the capacitance-based sensors, the performance of the sensors was tested in fluids of known dielectric permittivity ($\epsilon_{r}$). The impedance-based ThetaProbe sensor was used as the secondary standard to contextualize the performance of the tested sensors in some of the experiments.

The overall value for money of the sensors is reflected in their precision performance, i.e., the precision performance, on average, of the sensors followed the order of SMEC300 > SM100 > YL69 > YL100, which was almost the same as that of sensor costs, particularly considering that the ThetaProbe sensor was highest in precision. The accuracy of the sensors, on average, followed the order of SM100 > SMEC300 > YL100 > YL69.

It was found that the low-cost capacitive sensors, with soil-specific calibration, can match the performance of the secondary standard and could possibly be used for irrigation management with ‘limited’ effects on irrigation efficiency (in the context of accuracy). Among the two capacitive sensors, the less-expensive SM100 sensor can be inferred as a more robust and field ready low-cost soil moisture sensor. This is due to its strong performance in fluids (which is a proxy to its measurement technique), consistent precision across soils, accurate performance particularly in the field soil, and reasonable sensitivity to variations in temperature and salinity conditions. The SMEC300 sensor was accurate (except in field silty-loam soil), more precise than the SM100 sensor, and was reasonable in its response to temperature and salinity variations. With its additional capabilities of measuring temperature and electrical conductivity (the results of which have been purposely left out in lieu of being out of the framework of soil ‘moisture’ sensor comparison), it also presents itself as a useful multipurpose low-cost sensor.

The resistive sensors perform well considering their price category. Both the sensors are less precise and less accurate than the capacitive sensors. The YL100 has limited accuracy and precision, particularly when operating in temperature sensitive conditions, and fails in varying salinity conditions. The YL69 sensor, on average, is as accurate but more precise than the YL100, and is additionally able to operate with expected response to variability in temperature and salinity (comparable even to the capacitive sensors), establishing it as a robust, very low-cost soil moisture sensor. Though neither of the resistive sensors can be recommended as a standalone soil moisture sensor for irrigation management solutions (due to their limited accuracy), they may be used in combination with more accurate soil moisture sensors to better characterize the spatiotemporal variability of field scale soil moisture.

Despite the limitations of the current experiments and having acknowledged the need for more comprehensive investigation (including field experiments), this study, which describes the laboratory performance evaluation and characterization of low and very low-cost soil moisture sensors, is a precondition for the realization of tangible progress within the larger framework of improving low-cost data-driven agricultural water management solutions.

## Figures and Tables

**Figure 1 sensors-20-00363-f001:**
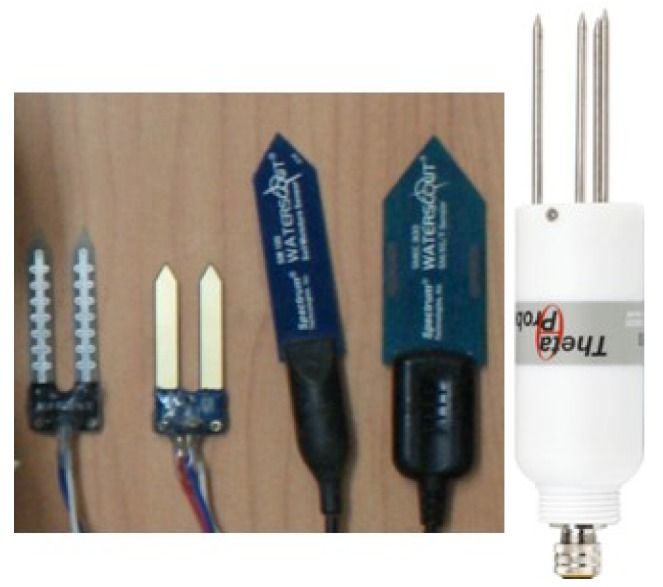
The four soil moisture sensors investigated in the study; from left to right (in the order of ascending cost): YL69, YL100, SM100, and SMEC300. The rightmost sensor is the secondary standard sensor, ThetaProbe.

**Figure 2 sensors-20-00363-f002:**
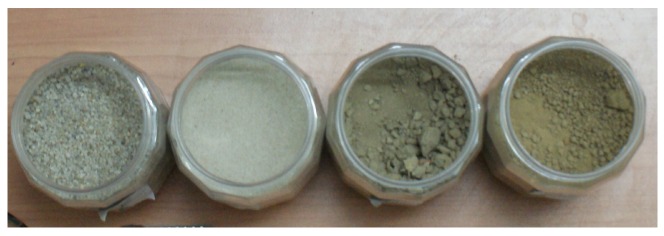
The four different soils used the study. From left to right: Soil 1: Grade I Sand [46], Soil 2: Grade III Sand [46], Soil 3: Silty-loam soil from local field, Soil 4: Graded Silty-loam soil.

**Figure 3 sensors-20-00363-f003:**
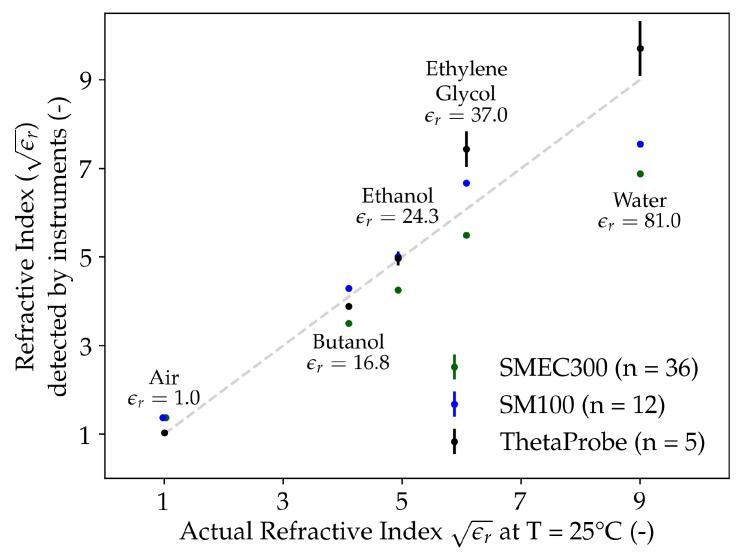
Response of the capacitive soil moisture sensors (SMEC300 and SM100) and secondary standard (impedance-based ThetaProbe) to fluids of known $\epsilon_{r}$ at 25 °C. The X- and Y-axis depict the actual and measured refractive indices ($\sqrt[{}]{\epsilon_{r}}$), respectively. Although the ThetaProbe measures $\sqrt[{}]{\epsilon_{r}}$ directly, the VWC values of SM100 and SMEC300 were converted to the corresponding $\sqrt[{}]{\epsilon_{r}}$ values based on the literature [10]. *n* is the total number of measurements in the experiment of a fluid, and the error bar shows the mean and standard error of the estimated values.

**Figure 4 sensors-20-00363-f004:**
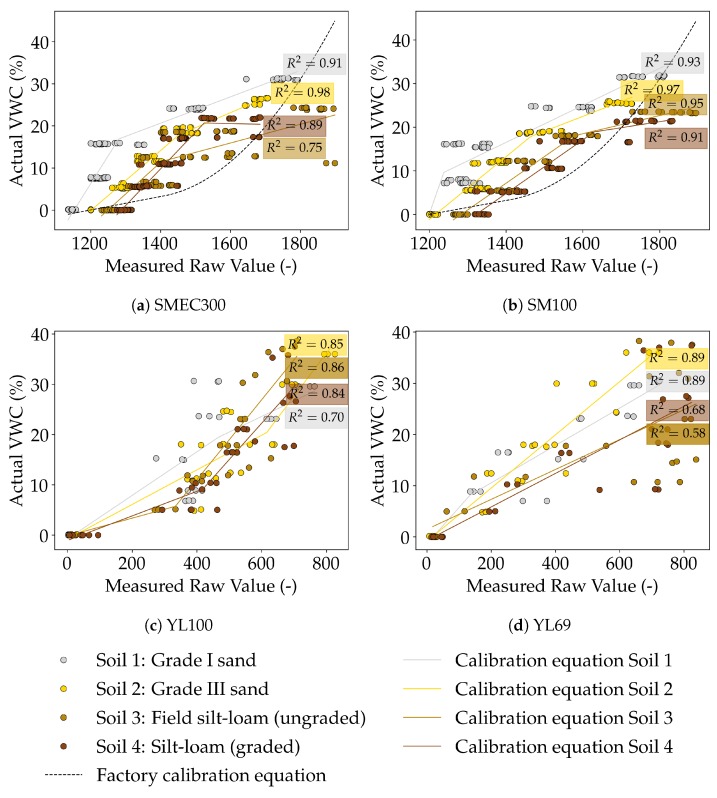
Calibration of capacitive sensors (**a**) SMEC300 and (**b**) SM100, and resistive sensors (**c**) YL100 and (**d**) YL69, in repacked soil using piecewise linear equations. The raw values correspond to either the raw readings from the Spectrum’s FieldScout reader (for capacitive sensors), or the raw outputs generated using the Arduino setup developed in-house (for resistive sensors). The coefficient of determination, *R*^2^, for each soil, is illustrated adjacent to the corresponding line.

**Figure 5 sensors-20-00363-f005:**
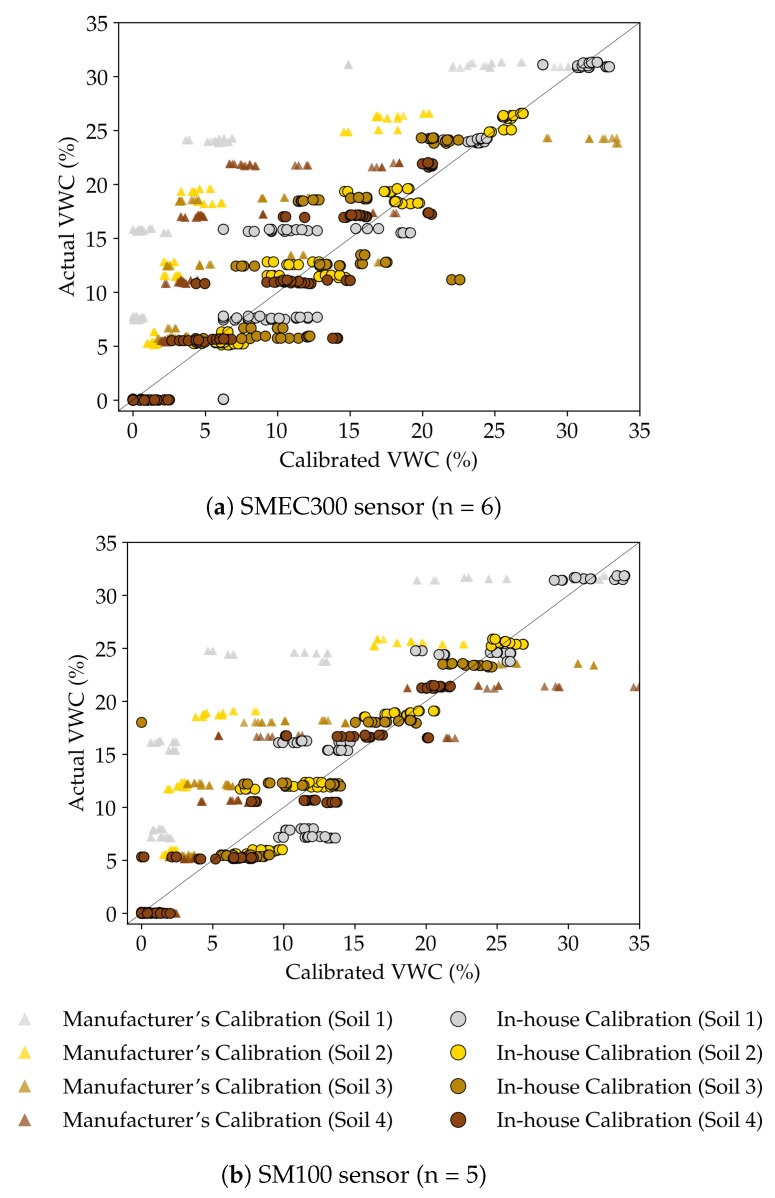
Comparison of manufacturer and in-house calibration equations for capacitive sensors (**a**) SMEC300 and (**b**) SM100 for the four different experimental soils.

**Figure 6 sensors-20-00363-f006:**
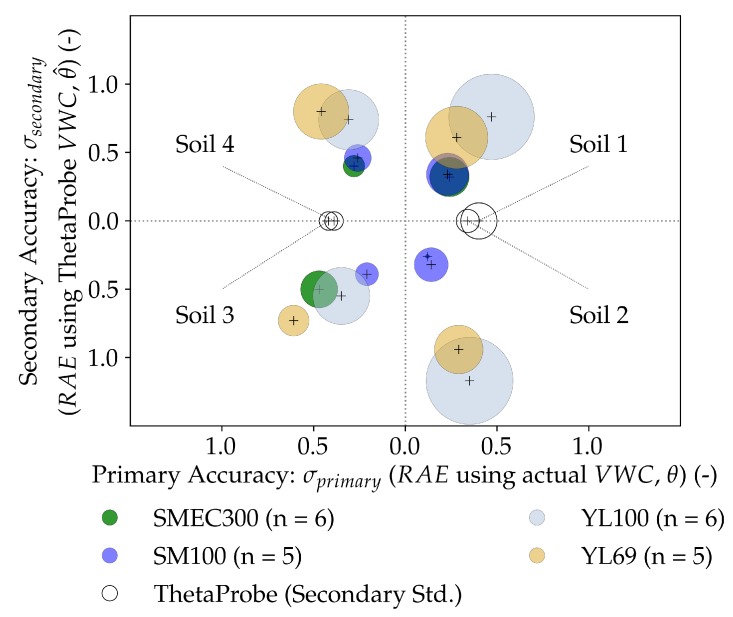
Accuracy (primary and secondary) and precision of different soil moisture sensors (SMEC300, SM100, YL69, YL100), in 4 different soils (corresponding to four quadrants). Overall accuracy, $\sigma_{eff}$ (Table 5), is the Euclidean distance of the bubble cross-hairs from the origin. The closer the bubble is to the origin, the more accurate the sensor is. Precision is indicated by the size of the bubbles ($radius=100\times s_{r,p}$); the smaller the bubble, the more precise the sensor. ‘n’ is the number of sensor units per sensor used in the experiment.

**Figure 7 sensors-20-00363-f007:**
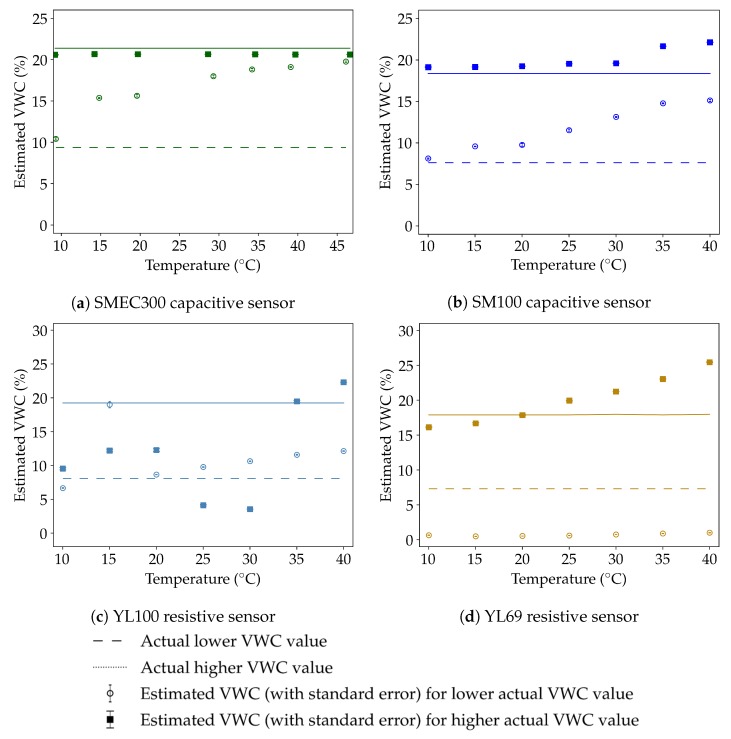
Temperature sensitivity of estimated VWC for different sensors: (**a**) capacitive SMEC300, (**b**) capacitive SM100, (**c**) resistive YL100, and (**d**) resistive YL69. The horizontal lines represent the actual VWC according to the legend. The hollow circular and solid square markers, along with their
error bars, represent the average and standard deviations of the calibrated/estimated sensor readings corresponding to the fixed lower and higher actual VWC values, respectively. Positive temperature effects are seen to different extents in all sensors, with the resistive sensors’ performance being limited by relatively lower accuracy and precision.

**Figure 8 sensors-20-00363-f008:**
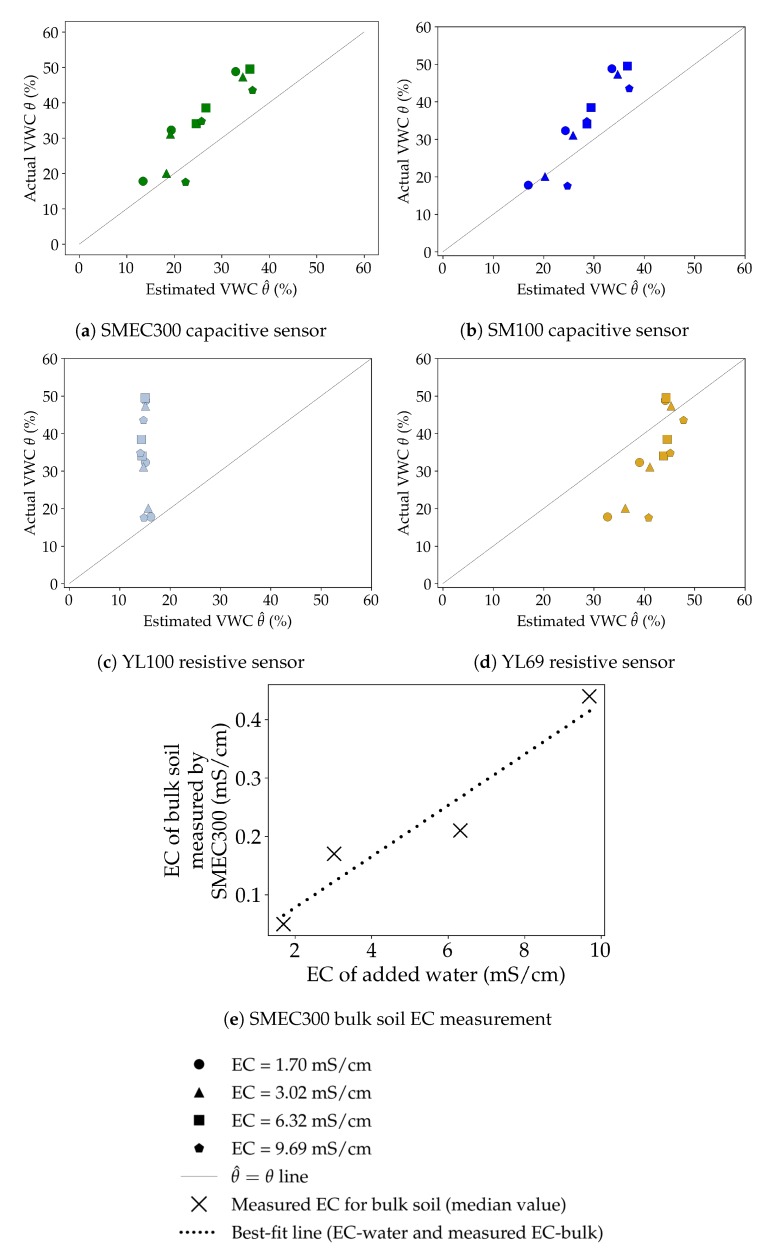
Effect of water of different electrical conductivity (EC) values on VWC measured (${\hat{\theta }}_{i}$) by different sensors: (**a**) capacitive SMEC300, (**b**) capacitive SM100, (**c**) resistive YL100, and (**d**) resistive YL69. (**e**) shows the relationship between the median values of the bulk soil EC measured by SMEC300 and the EC of water (with the corresponding best-fit line).

**Table 1 sensors-20-00363-t001:** Description of sensors used in the study.

Measurement Technique	Soil Moisture Sensor (Company)	Price (Quotation)	Nomenclature Used in Study
Capacitance based	SMEC300 Soil Moisture, Temperature and EC sensor (Spectrum Technologies)	$219.00	Low-cost *.
	SM100 Soil Moisture sensor (Spectrum Technologies)	$89.00	Low-cost.
Resistance based	YL100 Soil Hygrometer Detection Module soil moisture sensor (Electronicfans)	$3.89	Very Low-cost.
	YL69 Generic Soil Moisture Sensor Module (Kitsguru)	$2.11	Very Low-cost.
Impedance based	ThetaProbe ML3 Soil Moisture sensor (Delta-T Devices)	$516.33	High-cost, ‘true’ secondary standard sensor.

* Considering the additional temperature and EC sensing capabilities.

**Table 2 sensors-20-00363-t002:** Description of physical properties of the 4 soils used in the study [47].

Nomenclature Used in Study	Soil Description	Bulk Density [g/cc]	Soil Texture Classification
Soil 1	Grade I sand (1–2 mm)	1.82	Sand
Soil 2	Grade III sand (0.09–0.5 mm)	1.59	Sand
Soil 3	Field soil from experimental site at IIT Kanpur (Kanpur, India)	1.23	Silty-Loam
Soil 4	Graded Silty-Loam	1.20	Silty-Loam

**Table 3 sensors-20-00363-t003:** Fluids of known relative permittivity ($\epsilon_{r}$) used in the study.

Fluid	$\epsilon_{r}$ at T = 25 °C [2]
Air	1.0
Butanol	16.8
Ethanol	24.3
Ethylene-glycol	37.0
De-ionized water (Water)	81.0

**Table 4 sensors-20-00363-t004:** Electrical conductivities (EC) of the water samples and corresponding VWC measurements of the soil samples investigated in the salinity experiment.

EC of the Water Added [mS/cm]	Actual *VWC* [%]	Symbolic Representation in Figure 8
1.7	17.8	Circle (○)
1.7	32.3	
1.7	48.81	
3.02	20.08	Triangle(△)
3.02	31.12	
3.02	47.32	
6.32	34.09	Square(□)
6.32	38.5	
6.32	49.53	
9.69	17.59	Pentagon(⬠)
9.69	34.8	
9.69	43.53	

**Table 5 sensors-20-00363-t005:** The list of performance measures used in the study: $\theta_{i}$ denotes an actual VWC value; $\hat{\theta_{i}}$ represents a raw value measured by the sensor; $\bar{\theta }$ is the average of the actual VWC values; $\bar{\hat{\theta }}$ is the average of the raw values measured by the sensor; R(x) is the rank of *x* and *n* is the number of data points used in the computation. *k*, $n_{k}$, *m* and $s_{k}$ are the index of the current series, number of measurements in series *k*, total number of series, and corresponding standard deviation of the series, respectively, and are used to compute $s_{r,p}$.

Performance Metric	Description/Equation	Range (Ideal Value)
Coefficient of Determination ($R^{2}$) [59]	$R^{2}=\frac{\left(n\sum_{i=1}^{n}\hat{\theta_{i}}\theta_{i}\right)-\left(\sum_{i=1}^{n}\hat{\theta_{i}}\right)\left(\sum_{i=1}^{n}\theta_{i}\right)}{\sqrt{n\sum_{i=1}^{n}{\left(\hat{\theta_{i}}\right)}^{2}-{\left(\sum_{i=1}^{n}\hat{\theta_{i}}\right)}^{2}}\sqrt{n\sum_{i=1}^{n}{\left(\theta_{i}\right)}^{2}-{\left(\sum_{i=1}^{n}\theta_{i}\right)}^{2}}}$	0 to 1 (1)
Mean Absolute Error (MAE) [60]	$MAE=(\sum_{i=1}^{n}|\theta_{i}-\hat{\theta_{i}}|)/n$	0 to *∞* (0)
Pooled relative standard deviation ($s_{r,p}$) [54]	$s_{r,p}=\sqrt{\frac{\sum_{k=1}^{m}\left(n_{k}-1\right)s_{k}^{2}\left(1/{\bar{\hat{\theta_{k}}}}^{2}\right)}{\sum_{k=1}^{m}\left(n_{k}-1\right)}}$	0 to *∞* (0)
Relative Absolute Error (RAE) [60]	$RAE=\sum_{i=1}^{n}|\theta_{i}-\hat{\theta_{i}}|/\sum_{i=1}^{n}\left|\hat{\theta_{i}}-\bar{\theta }\right|$	0 to *∞* (0)
Root Mean Squared Error (RMSE) [60]	$RMSE=\sqrt{(\sum_{i=1}^{n}{\left(\theta_{i}-\hat{\theta_{i}}\right)}^{2})/n}$	0 to *∞* (0)
$\sigma_{effective}$	$\sigma_{eff}=\sqrt{{\left(\sigma_{primary}\right)}^{2}+{\left(\sigma_{secondary}\right)}^{2}}$	0 to *∞* (0)
$\sigma_{primary}$	RAE between in-house calibrated and actual VWC value	0 to *∞* (0)
$\sigma_{secondary}$	RAE between in-house calibrated and ThetaProbe VWC value	0 to *∞* (0)
Spearman’s Rank Correlation Coefficient ($r_{s}$) [61]	$r_{s}=\frac{\frac{1}{n}\sum_{i=1}^{n}\left(R\left(\hat{\theta_{i}}\right)-R\left(\bar{\hat{\theta }}\right)\right)\left(R\left(\theta_{i}\right)-R\left(\bar{\theta }\right)\right)}{\sqrt{\left(\frac{1}{n}\sum_{i=1}^{n}{\left(R\left(\hat{\theta_{i}}\right)-R\left(\bar{\hat{\theta }}\right)\right)}^{2}\right){\left(\frac{1}{n}\sum_{i=1}^{n}R\left(\theta_{i}\right)-R\left(\bar{\theta }\right)\right)}^{2}}}$	−1 to 1 (−1 or 1)

**Table 6 sensors-20-00363-t006:** Performance metrics of the capacitive (SMEC300 and SM100) and secondary standard (impedance-based ThetaProbe) sensors, in measuring refractive indices ($\sqrt[{}]{\epsilon_{r}}$) of fluids of known $\epsilon_{r}$ at 25 °C.

	SMEC300	SM100	ThetaProbe
MAE	0.87	0.55	0.48
RAE	0.22	0.27	0.24
RMSE	1.08	0.74	0.75
$s_{r,p}$	0.0062	0.0062	0.0405

**Table 7 sensors-20-00363-t007:** Spearman’s Rank Correlation Coefficient $r_{s}$ between the sensor readings and the actual soil volumetric water content (VWC) (θ) across the different soils. All the values are significant at α=5%.

	Low-Cost		Very Low-Cost	
	Capacitive Sensors		Resistive Sensors	
	**SMEC300**	**SM100**	**YL100**	**YL69**
**Soil 1**	0.93	0.92	0.78	0.91
**Soil 2**	0.96	0.97	0.89	0.94
**Soil 3**	0.84	0.94	0.94	0.73
**Soil 4**	0.95	0.92	0.94	0.85
**Average**	0.92	0.94	0.89	0.86

**Table 8 sensors-20-00363-t008:** Accuracy performance indicators of the tested sensors, with in-house calibration and manufacturer calibration (applicable only to capacitive sensors): Mean Absolute Error (MAE, in % VWC), Root Mean Squared Error (RMSE, in % VWC), and Relative Absolute Error (RAE or $\sigma_{primary}$, dimensionless). The same performance indicators are provided for the secondary standard sensor (for which no calibration equations were developed).

	Low-Cost Capacitive Sensors												Very Low-Cost Resistive Sensors						Secondary Standard		
	**SMEC300**						**SM100**						**YL100**			**YL69**			**ThetaProbe**		
	**Manufacturer Calibration**			**In-house Calibration**			**Manufacturer Calibration**			**In-house Calibration**			**In-house Calibration**			**In-house Calibration**			**Manufacturer Calibration**		
	**MAE**	**RMSE**	**RAE**	**MAE**	**RMSE**	**RAE**	**MAE**	**RMSE**	**RAE**	**MAE**	**RMSE**	**RAE**	**MAE**	**RMSE**	**RAE**	**MAE**	**RMSE**	**RAE**	**MAE**	**RMSE**	**RAE**
			$\sigma_{\mathrm{primary}}$			$\sigma_{\mathrm{primary}}$			$\sigma_{\mathrm{primary}}$			$\sigma_{\mathrm{primary}}$			$\sigma_{\mathrm{primary}}$			$\sigma_{\mathrm{primary}}$			$\sigma_{\mathrm{primary}}$
**Soil 1**	9.63	11.76	1.01	2.28	3.34	0.24	8.17	10.22	0.84	2.27	2.97	0.23	4.31	5.88	0.47	2.58	3.53	0.28	3.79	4.84	0.40
**Soil 2**	7.13	8.63	0.89	0.96	1.39	0.12	6.75	8.23	0.87	1.12	1.63	0.14	3.42	4.54	0.35	2.95	3.90	0.29	2.88	4.46	0.34
**Soil 3**	7.17	9.99	1.00	3.33	4.20	0.47	5.82	7.74	0.80	1.54	2.55	0.21	3.41	5.99	0.35	6.38	8.09	0.61	2.98	4.29	0.39
**Soil 4**	6.44	7.90	0.96	1.90	2.61	0.28	4.18	5.27	0.63	1.74	2.27	0.26	2.90	4.45	0.31	4.60	6.65	0.46	3.07	4.23	0.42
**Average**	7.59	9.57	0.97	2.12	2.88	0.28	6.23	7.86	0.78	1.67	2.36	0.21	3.51	5.21	0.37	4.13	5.54	0.41	3.18	4.45	0.39

**Table 9 sensors-20-00363-t009:** Comparison of precision performance of the tested sensors, based on pooled relative standard deviation, $s_{r,p}$ (% VWC). In-house calibration equations were used for the capacitive and resistive sensors, and Manufacturer calibration was used for the secondary standard sensor (for which no calibration equations were developed).

	Low-Cost		Very Low-Cost		Secondary
	Capacitive Sensors		Resistive Sensors		Standard
	**SMEC300**	**SM100**	**YL100**	**YL69**	**ThetaProbe**
**Soil 1**	0.51	0.55	1.11	0.81	0.47
**Soil 2**	0.05	0.44	1.13	0.63	0.30
**Soil 3**	0.48	0.30	0.74	0.40	0.24
**Soil 4**	0.28	0.35	0.78	0.72	0.24
**Average**	0.33	0.41	0.94	0.64	0.31

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
