# Peer review of "Laboratory Calibration and Performance Evaluation of Low-Cost Capacitive and Very Low-Cost Resistive Soil Moisture Sensors"

_sensors, 2020, doi:10.3390/s20020363_

Round 1

Reviewer 1 Report

This manuscript compared some properties among different sensors but lacked of innovation.

According to the article, the author had done some experiments about the sensor’s properties. But it doesn’t have any details about these experiments’ design and process in it.

Some experiment data can’t fit conclusion well. e.g. in line229, Figure3 indicates that SMEC300 was more accurate compared to the ThetaProbe in Ethylene Glycol and Water. But according to Figure3, SMEC300 was less accurate in water. So the data should be checked carefully.

Author Response

Response to Reviewer 1 Comments

Point 1: This manuscript compared some properties among different sensors but lacked of innovation.

Response 1: The laboratory characterization and testing of soil moisture sensors exists in the literature, but it is more prevalent for research-grade sensors, even in the context of low-cost sensors. The capabilities of sensors which belong to such low- and very-low cost categories for irrigation management are unknown, as studies characterizing their performance have not been conducted. In fact, the very-low cost (off-the-shelf) sensors do not even have available documentation. The novelty of this study is the characterization and testing of such sensors and their comparison with a secondary standard soil moisture sensor (a research grade relatively expensive sensor). We found that low-cost sensors with soil specific calibration can match the performance of the secondary standard and can be used for irrigation management. The very-low cost sensors have limited accuracy & precision and are sensitive to temperature and salinity variations. Hence, they alone may not be suitable for irrigation planning but in combination with low-cost sensors they may help in economical characterization of spatio-temporal variability of soil moisture. 

In addition, this study proposes a novel visual representation of multiple performance indices for multiple sensors and soils in a 2-D chart (Figure 6 of the revised manuscript). 

These two novel aspects have been explicitly stated in the introduction section (lines 87-91).

Point 2: According to the article, the author had done some experiments about the sensor’s properties. But it doesn’t have any details about these experiments’ design and process in it.

Response 2: The following experiments are conducted and their procedure and relevant references are provided in their respective sections. In addition, protocols followed in the experiments pertaining to Section 2.3.2 (Calibration of sensors with repacked soils) are described with photographs using a platform (protocols.io) for reproducible methods available at dx.doi.org/10.17504/protocols.io.swnefde  

(i) Calibration of capacitive sensors with fluids (Section 2.3.1.): The methodology was inspired partly by Kargas and Soulis (2012) and Kargas and Soulis (2019).

(ii) Calibration of sensors with repacked soils (Section 2.3.2.): The methodology is briefly mentioned in the section, and a detailed step-by-step description, is given on protocols.io [47] (the DOI has been converted to a link for more convenient access). The data obtained from these experiments were used for quantifying sensors’ accuracy (Section 2.4.1) and precision (Section 2.4.2), and comparing the in-house calibration equations with manufacturer provided calibration equations (Section 3.1.3.)

2.5.1. Temperature sensitivity: The details of the experiment have been described (with the relevant references) in the section. 

2.5.2. Salinity sensitivity: Salinity sensitivity is motivated by the methodology adopted in Kargas and Soulis (2012) and Kargas and Soulis (2019). 

Point 3: Some experiment data can’t fit conclusion well. e.g. in line229, Figure3 indicates that SMEC300 was more accurate compared to the ThetaProbe in Ethylene Glycol and Water. But according to Figure3, SMEC300 was less accurate in water. So the data should be checked carefully.

Response 3: Thank you for the correction! The data have been reanalyzed and the results have been modified. The respective changes are in lines 252-253: “Figure 3 indicates that the ThetaProbe sensor was more accurate than both the SMEC300 and SM100 sensors in all the fluids except Ethylene Glycol.” 

References (from authors):

Kargas, G., & Soulis, K. X. (2012). Performance analysis and calibration of a new low-cost capacitance soil moisture sensor. Journal of Irrigation and Drainage Engineering, 138(7), 632-641. Kargas, G., & Soulis, K. X. (2019). Performance evaluation of a recently developed soil water content, dielectric permittivity, and bulk electrical conductivity electromagnetic sensor. Agricultural water management, 213, 568-579.

Reviewer 2 Report

In my opinion, the study undertake the comprehensive analysis between the output of the low-very low-cost soil moisture sensor, including SM100, SMEC300, YL100 and YL69. In addition, the other analysis have been achieved. However, the creativity is not strong enough and the scientific meaning is not outstanding. Besides, the length is not long enough. So I can’t be convinced by the creativity to be published in ‘Sensors’ journal. I hope the author can enhance the scientific meaning or make the outstanding scientific phenomenon.

It is worth noting that my opinion is only for reference for I'm not familiar with such a area in the manuscript.

Overall, the manuscript is well written both in the grammar and structure. It’s quality is well. However, the scientific meaning of the study seems not high enough.

1. The conclusion concerning to the five questions in section 1 is not clear and not well clarified in section 3 or 4. So the conclusion or regular pattern should be highlighted and looks clear.

2. Though the analysis is clear and detailed. However, the sample size of involved four soil moisture sensors is not enough to reveal the objective facts.

3. I can’t find the link about protocols.io [45]. Please give the more detailed links.

4. The Indian Standard in Figure 2 should be given reference address.

Reviewer 3 Report

Review: Manuscript Number: sensors-667872

Laboratory Calibration and Performance Evaluation of Low-cost Capacitive and Very Low-cost Resistive Soil Moisture Sensors

This article reports studies which exploring the performance in laboratory conditions, of two low-cost capacitive sensors, SMEC300 and SM100 and two very low-cost resistive sensors, the Soil Hygrometer Detection Module Soil Moisture Sensor (YL100) by Electronicfans and the Generic Soil Moisture Sensor Module (YL69) by KitsGuru. Each sensor was calibrated in different repacked soils, and tested to evaluate accuracy, precision and sensitivity to variations in temperature and salinity. The capacitive sensors were additionally tested for their performance in liquids of known dielectric constants.

I don’t have a lot of comments, but maybe a central comment. Why did not examined the relationship between the actual VWC and square root of εr in all soils and sensors? The relationship between the actual VWC and square root of εr is linear in all cases? The occurrence of a linear equation between the square root of permittivity (εr) and actual VWC has been observed for several dielectric sensors, mainly in relation to inorganic soils (Seyfried et al. 2005;Kelleners et al., 2005; Kargas et al. 2011; Kargas and Soulis, 2019). However, the linear relationship parameters are significantly different between different soil types and sensors. It is better for the authors to present the data in the form actual VWC and square root of εr because a researcher could compare the differences between the soils and sensors.

Specific comments

l.219-220. Please insert the expression

l.221-223. It is better to presented in figure 3 the εr data

l.267. In this section you should presented the calibration curves and discuss the values of parameters of the linear relationship in each case (soils and sensors).

In the section (3.3.2. Salinity Sensitivity) you may present the bulk soil electrical conductivity values for SMEC300 sensors and the values of parameters for the linear relationship for each salinity level.

References

1)   Seyfried, M. S., Grant, L. E., Du, E., and Humes, K. (2005). “Dielectric loss and calibration of the Hydra Probe soil water sensor.” Vadose Zone J.,

4(4), 1070–1079.

2) Kargas, G., Kerkides, P., Seyfried, P. M., and Sgoumbopoulou, A.

(2011). “Wet sensor performance in organic and inorganic media with heterogeneous moisture distribution.” Soil Sci. Soc. Am. J., 75(4), 1244–1252.

3) Kelleners, T., Seyfried, M., Blonquist, J., Bilskie, J., Chandler, D., 2005.       Improved interpretation of water content reflectometer measurements in soils. Soil Sci. Soc. Am. J. 69, 1684–1690.

4) George Kargas, Konstantinos X. Soulis. Performance evaluation of a recently developed soil water content, dielectric permittivity, and bulk electrical conductivity electromagnetic sensor. https://doi.org/10.1016/j.agwat.2018.11.002.

Author Response

Note: Please see the attachment for major changes in the manuscript, which are highlighted.

Response to Reviewer 3 Comments

This article reports studies which exploring the performance in laboratory conditions, of two low-cost capacitive sensors, SMEC300 and SM100 and two very low-cost resistive sensors, the Soil Hygrometer Detection Module Soil Moisture Sensor (YL100) by Electronicfans and the Generic Soil Moisture Sensor Module (YL69) by KitsGuru. Each sensor was calibrated in different repacked soils, and tested to evaluate accuracy, precision and sensitivity to variations in temperature and salinity. The capacitive sensors were additionally tested for their performance in liquids of known dielectric constants.

Point 1: I don’t have a lot of comments, but maybe a central comment. Why did not examined the relationship between the actual VWC and square root of εr in all soils and sensors? The relationship between the actual VWC and square root of εr is linear in all cases? The occurrence of a linear equation between the square root of permittivity (εr) and actual VWC has been observed for several dielectric sensors, mainly in relation to inorganic soils (Seyfried et al.2005; Kelleners et al., 2005; Kargas et al. 2011; Kargas and Soulis, 2019). However, the linear relationship parameters are significantly different between different soil types and sensors. It is better for the authors to present the data in the form actual VWC and square root of εr because a researcher could compare the differences between the soils and sensors.

Response 1: The reasons for not exploring the relationship between actual VWC and square root of εr were different for the capacitive and the resistive sensors. 

The capacitive sensors (SM100 & SMEC300), manufactured by Spectrum Technologies Inc., output only Raw AD (analog-digital) values or VWC values (through manufacturer's calibration equation which is a function of Raw AD or Voltage Ratio) and do not provide εr values. Additional outputs for SMEC300 are temperature and Electrical Conductivity (EC). We had contacted the Soil/Water Product Manager of Spectrum Technologies (Mr. Doug Kieffer), who confirmed by email (September 5, 2018) that there is no way to access the εr values. The respective manuals are available online, for SMEC300: https://www.specmeters.com/assets/1/22/6470_SMEC3004.pdf and SM100: http://www.specmeters.com/assets/1/22/6460_SM100_(correct_Soilless-Web)1.pdf.

The resistive sensors (YL100 & YL69) respond to variation in resistance and not εr. This was also verified by dipping them in liquids of known εr at 25°C. Hence, there could not be any calibration equations developed between εr and actual VWC values. 

To summarize:

In this study, we explored the relationship between actual VWC and the sensor output (Raw AD value proportional to the voltage ratio). Linear, piecewise linear and polynomial equations were tested to explain the relationship between actual VWC and sensor output, to minimize the sum of squared residuals. The piecewise linear functions were found to best explain this relationship, and were hence chosen as the calibration equations. Such calibration equations relating actual VWC and sensor outputs (related to voltage) have been reported in previous publications: Czarnomski et al. (2005), Sakaki et al. (2008), Kinzli et al. (2012), Matula et al. (2016), etc.

Point 2: l.219-220. Please insert the expression

Response 2: Expression for Topp et al. (1980) has been included.

Point 3: l.221-223. It is better to presented in figure 3 the εr data

Response 3: εr values have been added to the labeling of the data points in Figure 3. 

Point 4: l.267. In this section you should presented the calibration curves and discuss the values of parameters of the linear relationship in each case (soils and sensors).

Response 4: Please refer to the response to point number 1 above. 

Point 5: In the section (3.3.2. Salinity Sensitivity) you may present the bulk soil electrical conductivity values for SMEC300 sensors and the values of parameters for the linear relationship for each salinity level.

Response 5: As suggested, a figure showing the bulk soil EC measured by SME300 and the variation between estimated and actual VWC is given in the figure below. Since a discussion on the bulk soil EC was not the focus of the study, it was not included in the manuscript. However, if the reviewer suggests, we can replace Figure 8 with the one given below.

References

1)   Seyfried, M. S., Grant, L. E., Du, E., and Humes, K. (2005). “Dielectric loss and calibration of the Hydra Probe soil water sensor.” Vadose Zone J.,

4(4), 1070–1079.

2) Kargas, G., Kerkides, P., Seyfried, P. M., and Sgoumbopoulou, A.

(2011). “Wet sensor performance in organic and inorganic media with heterogeneous moisture distribution.” Soil Sci. Soc. Am. J., 75(4), 1244–1252.

3) Kelleners, T., Seyfried, M., Blonquist, J., Bilskie, J., Chandler, D., 2005.       Improved interpretation of water content reflectometer measurements in soils. Soil Sci. Soc. Am. J. 69, 1684–1690.

4) George Kargas, Konstantinos X. Soulis. Performance evaluation of a recently developed soil water content, dielectric permittivity, and bulk electrical conductivity electromagnetic sensor. https://doi.org/10.1016/j.agwat.2018.11.002.

References (from authors)

Czarnomski, N. M., Moore, G. W., Pypker, T. G., Licata, J., & Bond, B. J. (2005). Precision and accuracy of three alternative instruments for measuring soil water content in two forest soils of the Pacific Northwest. Canadian journal of forest research, 35(8), 1867-1876. Kinzli, K. D., Manana, N., & Oad, R. (2011). Comparison of laboratory and field calibration of a soil-moisture capacitance probe for various soils. Journal of Irrigation and Drainage Engineering, 138(4), 310-321. Matula, S., Báťková, K., & Legese, W. (2016). Laboratory performance of five selected soil moisture sensors applying factory and own calibration equations for two soil media of different bulk density and salinity levels. Sensors, 16(11), 1912. Sakaki, T., Limsuwat, A., Smits, K. M., & Illangasekare, T. H. (2008). Empirical two‐point α‐mixing model for calibrating the ECH2O EC‐5 soil moisture sensor in sands. Water resources research, 44(4). Topp, G. C., Davis, J. L., & Annan, A. P. (1980). Electromagnetic determination of soil water content: Measurements in coaxial transmission lines. Water resources research, 16(3), 574-582.

Reviewer 4 Report

In this study two low cost EM soil moisture sensors and two very low cost resistive soil moisture sensors are evaluated experimentally in much detail. The study is interesting, very well written and the methodology used seems to be scientifically sound. The manuscript is also generally easy to read and understand except from the relatively long descriptions of the evaluations results (see my detailed comments).

My only hesitation is related with the presentation of the results related with the accuracy of the sensors and especially with the absence of any metrics that can provide the magnitude of the sensors errors in terms of soil water content. This is a very significant drawback of this study because the error expressed in terms of SWC is a really important information for the potential users in order to be able to evaluate if the sensors are appropriate for their application. For example, an SWC error of 0.05m3/m3 (or 5%) may seem small but as regards irrigation management is prohibiting because the range of available soil moisture in many soils can be even lower than 10%. Accordingly, I believe that RMSE or MAE or both metrics should be provided as well in all performance evaluations. Then the results should be also discussed in this context.

My second major comment is related to the final part of the discussion suggesting the use of very low cost sensors (and as I can see generally low accuracy) for irrigation management without considering the limitations and possible problems that can be caused due to erroneous SWC readings (e.g. excessive crops stress and production loss). I cannot also see how the use of multiple low accuracy sensors can solve this problem. Generally, errors in SWC measurments may have a significant effect in irrigation efficiency or crops yield (see for example Soulis K.X., Elmaloglou S., and Dercas N., 2015, Investigating the effects of soil moisture sensors positioning and accuracy on soil moisture based drip irrigation scheduling systems. Agricultural Water Management, 148, 258–268, doi: http://dx.doi.org/10.1016/j.agwat.2014.10.015 ). Accordingly, the discussion should also consider the accuracy of the SWC sensors expressed in SWC units. I believe that in general relevant suggestions of using low accuracy sensors in irrigation management should be given with great caution and after careful evaluation of the possible consequences.

Apart from the above comments I only have some other minor comments that included in the commented pdf file of the manuscript.

The above comments are important but I believe that it isn’t really difficult for the authors to address them based on the very good job that they have done in this study. Accordingly, I suggest a minor revision.

Author Response

NOTE: Please see the manuscript PDF file "adla-et-al_2019_changes-highlighted.pdf" for major changes which have been highlighted in yellow.

Response to Reviewer 4 Comments

In this study two low cost EM soil moisture sensors and two very low cost resistive soil moisture sensors are evaluated experimentally in much detail. The study is interesting, very well written and the methodology used seems to be scientifically sound. The manuscript is also generally easy to read and understand except from the relatively long descriptions of the evaluations results (see my detailed comments).

Point 1: My only hesitation is related with the presentation of the results related with the accuracy of the sensors and especially with the absence of any metrics that can provide the magnitude of the sensors errors in terms of soil water content. This is a very significant drawback of this study because the error expressed in terms of SWC is a really important information for the potential users in order to be able to evaluate if the sensors are appropriate for their application. For example, an SWC error of 0.05m3/m3 (or 5%) may seem small but as regards irrigation management is prohibiting because the range of available soil moisture in many soils can be even lower than 10%. Accordingly, I believe that RMSE or MAE or both metrics should be provided as well in all performance evaluations. Then the results should be also discussed in this context. 

Response 1: We wish to thank the reviewer for this input, and agree with the suggestion. Table 8 now includes MAE, RMSE and RAE (Relative Absolute Error) values for all the sensors and soil types that were tested in the study. Additionally, the accuracy performance of the manufacturer and in-house calibrations have been compared for the capacitive sensors (for which manufacturer calibration equations were available) using these three performance metrics (in particular, lines 332 onwards). Discussion based on these performance metrics has also been included in the revised manuscript. 

Point 2: My second major comment is related to the final part of the discussion suggesting the use of very low cost sensors (and as I can see generally low accuracy) for irrigation management without considering the limitations and possible problems that can be caused due to erroneous SWC readings (e.g. excessive crops stress and production loss). I cannot also see how the use of multiple low accuracy sensors can solve this problem. Generally, errors in SWC measurments may have a significant effect in irrigation efficiency or crops yield (see for example Soulis K.X., Elmaloglou S., and Dercas N., 2015, Investigating the effects of soil moisture sensors positioning and accuracy on soil moisture based drip irrigation scheduling systems. Agricultural Water Management, 148, 258–268, doi: http://dx.doi.org/10.1016/j.agwat.2014.10.015). Accordingly, the discussion should also consider the accuracy of the SWC sensors expressed in SWC units. I believe that in general relevant suggestions of using low accuracy sensors in irrigation management should be given with great caution and after careful evaluation of the possible consequences.

Response 2: 

The text in Section 3.4. (Further Discussion) has been changed to address the concern raised, using errors in SWC units. In particular, please note the following (lines 495-501): “The efficiency of soil moisture based irrigation scheduling systems is dependent strongly on the sensor accuracy, with 3% errors in soil moisture sensors possibly leading to ‘critical’ effects on irrigation efficiency (Soulis et al., 2015). Hence, while the capacitive sensors tested in this study had accuracy levels (<2% VWC on average) possibly leading to ‘limited’ effects on irrigation efficiency, using resistive sensors independently (with an accuracy of <4% VWC on average) could have potentially critical effects. The actual effect of these sensors on irrigation water use efficiency can be determined with comprehensive field experiments.
Soil moisture at field scale exhibits high spatial and temporal variability (Vereecken et al., 2014). Instead of a sparse network of capacitive sensors, a dense network combining capacitive and resistive sensors could help better characterize the spatio-temporal variability of soil moisture, which may potentially improve irrigation management. Such a characterization using a combination of low- and very-low cost soil moisture sensors has not been attempted, as per the knowledge of the authors. In Section 4. (Summary and Conclusions), lines 546-549 state: “Though neither of the resistive sensors can be recommended as a standalone soil moisture sensor for irrigation management (due to their limited accuracy), they may be used in combination with more accurate soil moisture sensors to better characterize the spatio-temporal variability of field scale soil moisture.”

Point 3: Apart from the above comments I only have some other minor comments that included in the commented pdf file of the manuscript.

Response 3: Specific points have been addressed in the subsequent points, as well as in the modified PDF file.

The above comments are important but I believe that it isn’t really difficult for the authors to address them based on the very good job that they have done in this study. Accordingly, I suggest a minor revision.

We thank the reviewer for the constructive comments. 

Comments from PDF file

Point 3a (line 229): I am not sure what do you mean? These values are respective to what?

Response 3a: Apologies for the typos. These were the MAE, RAE and RMSE values for the SM100 (given in Table 6) which were not properly removed from the textual description. The text has now been appropriately edited. 

Point 3b (Figure 4): Figure 4 is somehow confusing. I can also see that it could be probably easier (and more objective) to fit a single continuous higher order equation. Did you try it?

Response 3b: Yes, we tried linear, piecewise linear and polynomial (degree = 2 and 3) equations. Piecewise linear equations performed best in terms of visual fit and adjusted coefficient of determination, and hence were chosen as the calibration equations.

Point 3c (line 330 onwards): The absence of any metrics that can provide the magnitude of the sensors errors in terms of soil water content is a very significant drawback of this study. You should provide RMSE or MAE or both values as well (as presented in Table 3) in all performance evaluations. The error expressed in terms of SWC is really important in order for the potential users to be able to evaluate if these sensors are appropriate for their application. For example an error of 0.05m3/m3 (or 5%) may seem small but in terms of SWC for irrigation management is prohibiting as the range of available soil moisture in many soils can be even lower than 10%. 

Accordingly, I believe that this information should be added and discussed.

Response 3c: Please see response to Point 1 above.

Point 3d (line 383 onwards): Descriptions aren't bad but they are somehow tiresome to read. A table accompanied be a sorter description would be easier to read. 

The tables will also make it easier to present additional information such as RMSE and MAE. (Just a suggestion).

Response 3d: Table 8 has been added (with MAE, RMSE and RAE metrics for all sensors and soils). Table 9 is added to describe the precision performance of all the sensors. Moreover, the text has been shortened, particularly in Section 3.2 (Performance measures for the sensors). We hope that the revised manuscript is easier to read. 

Point 3e (line 485 onwards): This part of the discussion requires some further thinking. High errors in SWC measurements may result in very negative effects in terms of irrigation efficiency that could be worst than other alternatives or even empirical approaches. Even small errors may have a significant effect in irrigation efficiency (see  for example Soulis K.X., Elmaloglou S., and Dercas N., 2015, Investigating the effects of soil moisture sensors positioning and accuracy on soil moisture based drip irrigation scheduling systems. Agricultural Water Management, 148, 258–268, doi: http://dx.doi.org/10.1016/j.agwat.2014.10.015 ). 

Please also see my relative comment about the absence of information about the accuracy of the sensors in terms of SWC (e.g. RMSE, MAE).

I believe that the suggestions of using very low accuracy sensors in irrigation management should be given with great caution and after careful evaluation of the consequences.

Response 3e: Please see response to Point 2 above.

References (from authors):

González-Teruel, J. D., Torres-Sánchez, R., Blaya-Ros, P. J., Toledo-Moreo, A. B., Jiménez-Buendía, M., & Soto-Valles, F. (2019). Design and Calibration of a Low-Cost SDI-12 Soil Moisture Sensor. Sensors, 19(3), 491. Kinzli, K. D., Manana, N., & Oad, R. (2011). Comparison of laboratory and field calibration of a soil-moisture capacitance probe for various soils. Journal of Irrigation and Drainage Engineering, 138(4), 310-321. Vereecken, H., Huisman, J. A., Pachepsky, Y., Montzka, C., Van Der Kruk, J., Bogena, H., ... & Vanderborght, J. (2014). On the spatio-temporal dynamics of soil moisture at the field scale. Journal of Hydrology, 516, 76-96.

Round 2

Reviewer 1 Report

Most comments have been accepted by authors.

Author Response

Response to Reviewer 1 Comments (Round 2)

Point: Most comments have been accepted by authors. 

Response: Thank you very much. Please find the modified PDF file with changes highlighted in yellow.

Reviewer 2 Report

In my opinion, this paper is worth publishing in present form.

Author Response

Response to Reviewer 2 Comments (Round 2)

Point: In my opinion, this paper is worth publishing in present form.

Response: Thank you very much. Please find the modified PDF file with changes highlighted in yellow.

Reviewer 3 Report

I suggest to replace Figure 8 

Author Response

Response to Reviewer 3 Comments (Round 2)

Point 1: I suggest to replace Figure 8.

Response 1: Figure 8 (a-d) could not be completely replaced since a full comparison of all the sensors requires the results of each of the sensors to be presented. Instead, a sub-figure, Figure 8 (e) has been added, which shows the relationship between the bulk soil EC measured by SMEC300 and the EC of the added water (with the corresponding best-fit line). Additionally, the result is explained in lines 476-480.